# IBCircuit: Towards Holistic Circuit Discovery with Information Bottleneck

**Tian Bian** [* 1 2]  **Yifan Niu** [* 3]  **Chaohao Yuan** [1]  **Chengzhi Piao** [4]  **Bingzhe Wu** [5]  **Long-Kai Huang** [6]  **Yu Rong** [2 7]
**Tingyang Xu** [2 7]  **Hong Cheng** [1]  **Jia Li** [3]

## Abstract

Circuit discovery has recently attracted attention as a potential research direction to explain the non-trivial behaviors of language models. It aims to find the computational subgraphs, also known as *circuits*, within the model that are responsible for solving specific tasks. However, most existing studies overlook the holistic nature of these circuits and require designing specific corrupted activations for different tasks, which is inaccurate and inefficient. In this work, we propose an end-to-end approach based on the principle of Information Bottleneck, called IBCircuit, to identify informative circuits holistically. IBCircuit is an optimization framework for holistic circuit discovery and can be applied to any given task without tediously corrupted activation design. In both the Indirect Object Identification (IOI) and Greater-Than tasks, IBCircuit identifies more faithful and minimal circuits in terms of critical node components and edge components compared to recent related work. The code is available at https://github.com/ivanniu/IBCircuit.

## 1. Introduction

Circuit discovery in language models usually involves identifying the subgraphs (*circuits*) within the model that are responsible for solving specific tasks (Olah et al., 2020). Previous efforts to identify circuits within language models have led to the discovery of components (e.g., attention

---
[*]Equal contribution; alphabetical order by last name. [1]Department of Systems Engineering and Engineering Management, The Chinese University of Hong Kong, Hong Kong, China [2]DAMO Academy, Alibaba Group, Hangzhou, China [3]Hong Kong University of Science and Technology (Guangzhou), Guangzhou, China [4]Department of Computer Science, Hong Kong Baptist University, Hong Kong, China [5]School of Artificial Intelligence, Shenzhen University, Shenzhen, China [6]Tencent, Shenzhen, China [7]Hupan Lab, Hangzhou, China. Correspondence to: Tingyang Xu <xuty_007@hotmail.com>, Jia Li <jialee@ust.hk>.

*Proceedings of the 42$^{nd}$ International Conference on Machine Learning*, Vancouver, Canada. PMLR 267, 2025. Copyright 2025 by the author(s).

heads and MLPs in Transformers (Vaswani et al., 2017)) that either partially or fully explain the model's behaviors on tasks like Indirect Object Identification (IOI), modular arithmetic, and forecasting subsequent dates (Wang et al., 2022; Nanda et al., 2023; Hanna et al., 2024). The challenge of circuit discovery lies in the fact that circuits are hidden in the complex black boxes of language models which involve complex non-linear interactions in densely-connected layers and embed in a high-dimensional space (Wang et al., 2022; Li et al., 2025a).

Recent circuit discovery works seek to decode these models by reverse engineering (Räuker et al., 2023), such as activation patching (Meng et al., 2022) and attribution patching (Nanda, 2023). The activation patching (Geiger et al., 2021; Goldowsky-Dill et al., 2023; Conmy et al., 2023; Wang et al., 2022) performs causal interventions independently on each component, ignoring that the circuit is a holistic system rather than an independent combination. Additionally, they do not scale well with the model size. Another research line has proposed attribution patching (Nanda, 2023) and its variants (Chintam et al., 2023; Hanna et al., 2022; Marks et al., 2024) to efficiently estimate the importance of each component in the computational graph based on gradient methods. However, they still need to redesign corrupted activations for different tasks, which is inconvenient and complicated.

An ideal minimal circuit should (i) contain only the essential components for specific tasks, and (ii) exclude any irrelevant or redundant components. Intuitively, to identify high-quality circuits, we should maximize the circuit's relevance to the tasks while minimizing irrelevant components. Interestingly, the objective of circuit discovery aligns with the Information Bottleneck (IB) principle. The IB leverages Shannon mutual information to distill the compressed yet informative data distributions (Tishby et al., 2000). From the IB perspective, circuit discovery is to (i) find circuits that are the most informative to the specific tasks and (ii) compress them from the overall language models.

In this work, we propose the Information Bottleneck Circuit (IBCircuit), a novel IB framework designed to identify the informative components within Transformer-based language models. IBCircuit is an optimization framework for holistic

circuit discovery and can be applied to any given task without tediously corrupted activation design. To parameterize a circuit, IBCircuit injects controllable Gaussian noise with learnable *IB weights* into various model components (e.g., the activations of attention heads and MLPs). It modulates the flow of clean information from the original pretrained model to its distorted version. The IBCircuit objective encourages the distorted flow to retain its informativeness. Therefore, IBCircuit preserves the most informative components with minimal noise injection. This process simulates information compression, ensuring that the most informative components are preserved while irrelevant components are filtered out. Finally, the circuit is formed by discretizing the continuous weights to select the most informative components. The contributions of this work are as follows:

- We propose a circuit discovery framework, IBCircuit, which utilizes information bottleneck to holistically identify the most informative and compressed circuit within the language models.

- We introduce a circuit parameterization strategy that incorporates noise injection with learnable *IB weights*, providing a task-agnostic alternative to the task-specific corrupted activation construction.

- We conducted experiments in the Indirect Object Identification (IOI) and Greater-Than tasks, verifying that IBCircuit can identify more faithful and minimal circuits in terms of selecting critical node and edge components compared to baseline methods.

## 2. Related Work

**Circuit Analysis**    Recent advances in circuit analysis focus on reverse-engineering neural networks through two primary methodologies: activation patching and attribution patching. The *activation patching* paradigm (Vig et al., 2020; Finlayson et al., 2021; Geiger et al., 2021; Goldowsky-Dill et al., 2023; Conmy et al., 2023; Wang et al., 2022) conducts causal interventions by modifying individual components' activations. Common implementations include overwriting activations with zeros (Cammarata et al., 2021; Olsson et al., 2022), substituting with dataset mean values (Wang et al., 2022; Hanna et al., 2024), or applying interchange interventions (Geiger et al., 2021; Wu et al., 2024). However, these approaches face two fundamental limitations: (1) They treat circuit components as independent entities, disrupting the model's holistic computation flow (Räuker et al., 2023); (2) The substituted activations often deviate from plausible activation distributions (Chan et al., 2022), and their sequential intervention process becomes computationally expensive for large models. In contrast, *attribution patching* methods (Nanda, 2023) and their variants (Chintam et al., 2023; Hanna et al., 2022; Marks et al.,

2024) employ gradient-based approaches to estimate component importance across the computational graph. While these techniques improve efficiency through parallelizable gradient computations, they introduce new practical challenges. Current implementations require task-specific engineering of corrupted activation patterns (Nanda, 2023), creating implementation complexity and limiting generalizability across different behaviors. This fundamental tension between intervention fidelity and computational efficiency remains unresolved in existing literature.

**Information Bottleneck**    The principle of the information bottleneck (IB) aims to extract a compressed yet predictive code from the input signal (Tishby et al., 2000; Yu et al., 2024). Alemi et al. (2017) initially introduced the variational information bottleneck (VIB) to deep learning interpretability research. Currently, IB and VIB primarily focus on informative representation learning and feature selection. In representation learning, researchers aim to learn a compressed representation with the IB principle (Luo et al., 2019; Qian et al., 2020; Wu et al., 2020; Liu et al., 2024). For feature selection, IB is used to select a subset of input features, such as image pixels or vector dimensions (Achille & Soatto, 2018b; Kim et al., 2021; Schulz et al., 2020). For instance, Yu et al. (2020) *et al.* present a graph information bottleneck to identify important subgraphs. Wu & Deng (2023) *et al.* introduce the Two-Stream Information Bottleneck (TIB) method, which uses a standard IB and a Reverse Information Bottleneck (RIB) to detect unknown objects.

## 3. Preliminaries

**Neural Circuits**    A Neural Circuit for a given task is the minimal computation subgraph $\mathcal{C} \subset \mathcal{G}$, where $\mathcal{C}$ and $\mathcal{G}$ denote the set of components in the circuit and complete model, respectively (Olah et al., 2020). In the computational graph of transformer-based language models $\mathcal{G}$, the nodes are defined as MLPs and attention heads, and the edges are defined as the dependencies between nodes (Li et al., 2025b; Chang et al., 2024). The objective of circuit discovery is to identify a sparse subgraph that encapsulates the behavior of the complete model for a specific task. We denote the output of the circuit, given the original and distorted information flows $x, \tilde{x}$, as $p_{\mathcal{C}}(y \mid x, \tilde{x})$, while the output of the complete model is represented as $p_{\mathcal{G}}(y \mid x)$. Formally, the goal of circuit discovery can be expressed as follows:

$$
\begin{aligned}
\arg\min_{\mathcal{C}} \quad & \mathbb{E}_{(x,\tilde{x}) \in \mathcal{T}} \left[ D(p_{\mathcal{G}}(y \mid x) \,\|\, p_{\mathcal{C}}(y \mid x, \tilde{x})) \right], \\
\text{s.t. } & 1 - |\mathcal{C}|/|\mathcal{G}| \geq c
\end{aligned}
\tag{1}
$$

The constraint $c$ is designed to ensure a desired sparsity for the circuit. The set $\mathcal{T}$ represents the relevant task distribution. The objective function $D$ captures the discrepancy between the outputs of the complete model and the circuit.

**Information Bottleneck** The Information Bottleneck (IB) (Tishby et al., 2000) is to optimize the representation $Z$ to capture the minimal sufficient information within the input data $\mathcal{D}$ to predict the target $Y$. Its objective can be formulated as follows:

$$\min_{\mathbb{P}(Z|\mathcal{D}) \in \Omega} \text{IB}_\beta(\mathcal{D}, Y; Z) \triangleq [-I(Y; Z) + \beta I(\mathcal{D}; Z)], \quad (2)$$

where $\Omega$ defines the search space of the optimal model, and $I(\cdot; \cdot)$ denotes the mutual information (Cover, 1999). The first term $-I(Y; Z)$ encourages $Z$ to be informative about the target $Y$, while the second term $I(\mathcal{D}; Z)$ ensures that $Z$ does not receive irrelevant information from $\mathcal{D}$. The IB provides a critical principle for representation learning: an optimal representation should contain the minimal sufficient information for the downstream prediction task.

# 4. IBCircuit

## 4.1. Intuition: Informative Circuit

In language model, let us consider the specific task $X = \{x^{(i)}\}_{i=1}^N$ consisting of $N$ i.i.d. samples and the target $Y = \{y^{(i)}\}_{i=1}^N$. We assume that the data are generated by some random process, involving an unobserved random circuit $\mathcal{C} = \{c^{(i)}\}_{i=1}^N$. An ideal minimal circuit (i) contains the most informative components for specific tasks, and (ii) does not include irrelevant or redundant components. Intuitively, the goal of circuit discovery in Eq. (1) aligns with the concept of the Information Bottleneck (IB) principle. We denote $\mathcal{G} = \{g^{(i)}\}_{i=1}^N$ as the whole computation graph of the transformer-based model, $Y$ as the output of the model on specific tasks $X$, and $\mathcal{C}$ as the circuit composed of critical components. We reformulate Eq. (2) to obtain the objective of IBCircuit as follows:

$$\min_{\mathbb{P}(\mathcal{C}|\mathcal{G})} \text{IB}_\beta(\mathcal{G}, Y; \mathcal{C}) \triangleq [-I(Y; \mathcal{C}) + \beta I(\mathcal{G}; \mathcal{C})], \quad (3)$$

The first term encourages the circuit $\mathcal{C}$ to be informative on the targets $Y$. The second term ensures that $\mathcal{C}$ receives limited information from the whole computation graph $\mathcal{G}$, i.e., minimizing task-irrelevant components. However, in practice, the mutual information in Eq. (3) is intractable.

## 4.2. Estimation of Mutual Information

Exact computation of $I(Y; \mathcal{C})$ and $I(\mathcal{G}; \mathcal{C})$ is intractable. Hence, we introduce variational approximation to estimate the variational bounds on these two terms and propose the IBCircuit framework to find informative circuits. The derivation is detailed in Appendix A.

**Maximizing $I(Y; \mathcal{C})$.** We first examine the first term $I(Y; \mathcal{C})$ in Eq. (3), which encourages $\mathcal{C}$ to be informative of the output $Y$. We derive the variational lower bound of $I(Y; \mathcal{C})$ as follows:

**Proposition 4.1** (Variational lower bound of $I(Y; \mathcal{C})$)**.** *For the output $Y$ of the original transformer language model $\mathcal{G}$, the output $Y_\mathcal{C}$ of the given circuit $\mathcal{C}$, the variational lower bound can be written as:*

$$I(Y; \mathcal{C}) \geq -\frac{1}{N} \sum_{i=1}^N [D_{KL}(y^{(i)} || y_c^{(i)}) + H(y^{(i)})]. \quad (4)$$

*where $D_{KL}(\cdot || \cdot)$ is the Kullback-Leibler Divergence, $H(\cdot)$ is the entropy.*

In general, given a language model on a specific task, the $H(y^{(i)})$ is a constant. Therefore, we can omit this term, and the lower bound of Eq. (4) can be reformulated as $-\frac{1}{N} \sum_{i=1}^N D_{KL}(y^{(i)} || y_c^{(i)})$. This inequality is highly intuitive, demonstrating that maximizing $I(Y; \mathcal{C})$ can be achieved by minimizing the KL Divergence of the output between the whole computation graph and the circuit. In other words, the circuit contains the most informative components for specific tasks.

**Minimizing $I(\mathcal{G}; \mathcal{C})$.** For the second term $I(\mathcal{G}; \mathcal{C})$ in Eq. (3), which ensures $\mathcal{C}$ contains minimal irrelevant or redundant information about $\mathcal{G}$. Its variational upper bound is as follows:

**Proposition 4.2** (Variational upper bound of $I(\mathcal{G}; \mathcal{C})$)**.** *For the complete transformer language model $\mathcal{G}$, and the circuit $\mathcal{C}$, the variational upper bound can be written as:*

$$I(\mathcal{G}; \mathcal{C}) \leq D_{KL}[q(\mathcal{C}|\mathcal{G}) || p(\mathcal{C})], \quad (5)$$

where $q(\mathcal{C}|\mathcal{G})$ and $p(\mathcal{C})$ is the posterior and prior of the circuit. Minimizing the upper bound of $I(\mathcal{G}; \mathcal{C})$ constrains the irrelevant information that $\mathcal{C}$ retains from $\mathcal{G}$. Therefore, we can select the circuit $\mathcal{C}$ from $\mathcal{G}$ after training the IBCircuit with the tractable bounds.

## 4.3. Circuit Parameterization

To implement IBCircuit, we follow the variational method (Rockafellar & Wets, 2009) and assume the prior of the circuit to be an isotropic Gaussian. Given a pretrained model composed of $n$ components with intermediate activations $\mathbf{h} = [h_1, h_2, \cdots, h_n]$, we introduce *IB weights* $\lambda = [\lambda_1, \lambda_2, \cdots]$ to control information flow through $\lambda_i = \text{Sigmoid}(\omega_i)$ with learnable parameter $\omega_i$. To construct circuit, we adopt Gaussian noise $\epsilon_i \sim \mathcal{N}(\mu_i, \sigma_i^2)$ scaled by $\lambda_i \in (0, 1)$ to perturb activations, where $\mu_i$ and $\sigma_i^2$ are computed from batch activations of $h_i$. Following established circuit analysis protocols (Wang et al., 2022; Conmy et al., 2023), we parameterize the circuit on node and edge level:

**Node-wise Parameterization.** Following previous work on node-level circuit discovery (Wang et al., 2022), we focus

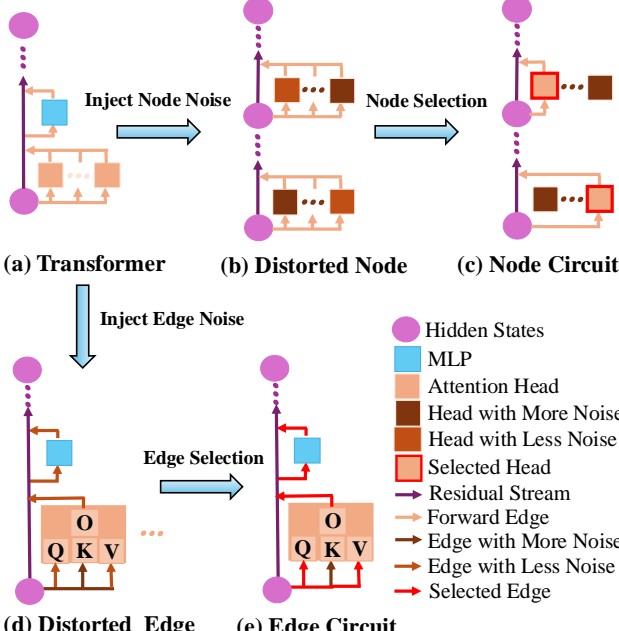

*Figure 1.* (a) Transformer blocks from the perspective of the residual stream. (b) Adding Gaussian noise to the activations of attention heads using node-wise *IB weights* and optimizing through the Information Bottleneck. (c) Selecting attention heads with less noise as the node-level circuit. (d) Adding Gaussian noise to the activations of source nodes using edge-wise *IB weights* and optimizing through the Information Bottleneck. (e) Selecting edges with less noise as the edge-level circuit.

on attention heads as node components. We perturb the output activations of each attention head in the Transformer architecture and train the distorted model holistically. For individual attention heads:

$$\hat{h}_i = \underbrace{\lambda_i h_i}_{\text{signal preservation}} + \underbrace{(1 - \lambda_i)\epsilon_i}_{\text{noise injection}}, \qquad (6)$$

where $h_i$ denotes the $i$-th attention head's activation and $\hat{h}_i$ denotes the distorted activation, $\lambda_i$ is node-wise *IB weight*.

**Edge-wise Parameterization.** Inspired by ACDC (Conmy et al., 2023), we explicitly model the Transformer's residual stream through two node types: (1) **Source Nodes**: Token/positional embeddings, attention head outputs, and MLP outputs. (2) **Target Nodes**: Q/K/V projection inputs and MLP inputs. Following the layer-wise residual architecture, each target node at layer $l$ (denoted as trg($l$)) aggregates information from *all preceding source nodes* up to layer $l - 1$ (denoted as src($< l$)), forming directed edges $\mathcal{E} = \{(j, i)|j \in \text{src}(< l), i \in \text{trg}(l)\}$. We parameterize the edge through:

$$\hat{h}_i = \sum_{(j,i)\in\mathcal{E}} \Big[ \underbrace{\lambda_{ji} h_j}_{\text{signal propagation}} + \underbrace{(1 - \lambda_{ji})\epsilon_j}_{\text{noise injection}} \Big], \qquad (7)$$

where $h_j$ denotes the $j$-th source node's activation and $\hat{h}_i$ denotes the distorted target node's activation, and $\lambda_{ji}$ denotes edge-wise *IB weight*.

The IB weights $\lambda$ implement differentiable information gates: $\lambda_i \to 1$ preserves original activations while $\lambda_i \to 0$ induces maximal noise. Crucially, our formulation enables *simultaneous optimization* of all $\lambda$ through holistic gradient flow across the computational graph and adaptive noise calibration via batch statistics. This contrasts with existing patching methods that require sequential intervention.

### 4.4. Objective for Training

The circuit $\mathcal{C}$ is designed to extract information from the pre-trained model $\mathcal{G}$ by approaching the original outputs $Y$. Our framework operates through two complementary mechanisms: (1) maximizing the mutual information between $\mathcal{C}$'s outputs and the target outputs $Y$ to maintain functional consistency, and (2) introducing controlled noise injection to minimize dependence on the pre-trained model $\mathcal{G}$, thereby preserving task-critical components while perturbing non-essential components. Considering the theoretical results in Propositions 4.1 and 4.2, the variational upper bound IBCircuit$_\beta(\mathcal{G}, Y; \mathcal{C})$ can be formulated as:

$$\frac{1}{N}\sum_{i=1}^{N} D_{KL}(y^{(i)}||y_c^{(i)}) + \beta D_{KL}[q(\mathcal{C}|\mathcal{G})||p(\mathcal{C})], \qquad (8)$$

where $\beta$ is the hyperparameter. Consider our circuit parameterization, the second term $D_{KL}[q(\mathcal{C}|\mathcal{G})||p(\mathcal{C})]$ can be further specified as:

$$D_{KL}[q(\mathcal{C}|\mathcal{G})||p(\mathcal{C})] = -\frac{\log A}{n} + \frac{A^2 + B^2 - 1}{2n}, \qquad (9)$$

where $A = -\sum_{i=1}^{n}(1 - \lambda_i)$, $B = \sum_{i=1}^{n}\frac{\lambda_i(h_i - \mu_i)}{\sigma_i}$, and $n$ is the number of component in the language model. Although the exact computation of $I(Y; \mathcal{C})$ and $I(\mathcal{G}; \mathcal{C})$ is intractable, we can calculate their variational bounds. Therefore, we adopt IBCircuit$_\beta(\mathcal{G}, Y; \mathcal{C})$ as the overall objective function. Compared to existing methods, (1) IBcircuit is an optimization framework that allows us to optimize the IB weight holistically through end-to-end training; (2) IBCircuit uses the original output $Y$ of the transformer model as supervision signals, avoiding the need to manually design corrupted activations.

### 4.5. Circuit Formation

The learned IB weights $\lambda$ encode component importance through their information transmission capacity. We develop circuit formation protocols for node and edge components:

**Node Component Selection.** Corresponding to node-wise perturbation in Eq. (6), we identify critical attention heads:

$$\mathcal{C}_{node}^* = \Big\{ i \in [n] \,\Big|\, \lambda_i > \tau_{\text{node}} \Big\}, \qquad (10)$$

where the adaptive threshold $\tau_{\text{node}}$ is determined via:

$$\tau_{\text{node}} = \inf \left\{ \tau \in (0,1) \,\Big|\, \sum_{i=1}^{n} \mathbb{I}(\lambda_i > \tau) \leq k_{\text{node}} \right\}, \quad (11)$$

with $k_{\text{node}}$ controlling the maximum allowed nodes.

**Edge Component Selection.** Similar to node component selection, we select essential residual connections through:

$$\mathcal{C}_{edge}^* = \left\{ (j,i) \in \mathcal{E} \,\Big|\, \lambda_{ji} > \tau_{\text{edge}} \right\}. \quad (12)$$

The edge threshold $\tau_{\text{edge}}$ is determined via:

$$\tau_{\text{edge}} = \inf \left\{ \tau \in (0,1) \,\Big|\, \sum_{i=1}^{|\mathcal{E}|} \mathbb{I}(\lambda_{ji} > \tau) \leq k_{\text{edge}} \right\}, \quad (13)$$

with $k_{\text{edge}}$ controlling the edge sparsity.

## 5. Experiments

In this section, we conduct the evaluation to answer the following research questions (RQs):

- **RQ1-Grounded in Previous Work**: Can IBCircuit effectively reproduce circuits taken from previous works that found the circuit explaining behavior for tasks?

- **RQ2-Ablation Study**: Are both KL loss and MI loss used for training IBCircuit necessary? How does the different $\alpha$ affect the effectiveness of IBCircuit? What is the contribution of each component in the IBCircuit?

- **RQ3-Faithfulness & Minimality**: Does IBCircuit avoid including components that do not participate in the behavior while maintaining better faithfulness?

- **RQ4-Scalability to Large Models**: Does IBCircuit have the scalability to be applied on large models?

### 5.1. Experiment Setting

**Tasks** We primarily focus on GPT-2 (Radford et al., 2019) for better evaluation, as it is a classical model typically studied from a circuit's perspective. We intentionally choose two tasks (IOI and Greater-Than) that have been studied before for fair comparison with previous work.

- **Indirect Object Identification (IOI)** (Wang et al., 2022): An IOI sentence involves an initial dependent clause, e.g., "When Mary and John went to the store", followed by a main clause, e.g., "John gave a drink to Mary." In this case, the indirect object (IO) is "Mary" and the subject (S) is "John". The IOI task is to predict the final token in the sentence to be the IO. IOI Circuit Discovery aims to identify which components of the model are crucial for performing such IOI tasks.

- **Greater-Than** (Hanna et al., 2024): In the Greater-Than task, models receive input like "The war lasted from the year 1741 to the year 17", and must predict a valid two-digit end year, i.e., one that is greater than 41. In this paper, we aim to identify which components of the model are crucial for predicting the end year.

**Baselines.** We compare the proposed method with the following methods designed for node component selection and edge component selection, respectively:

**Node Component Selection Methods:**

- **Subnetwork Probing (SP)** (Cao et al., 2021): SP learns a mask for each node in the circuit to determine if it is part of the circuit via gradient descent.

- **Automated Circuit DisCovery (ACDC)** (Conmy et al., 2023): ACDC is originally designed for edge component selection. We derive the score of a node by summarizing the impact of its removal on the model.

- **Attribution Patching (AP)** (Nanda, 2023): AP assigns scores to all nodes at the same time by leveraging gradient information and again prunes nodes below a certain threshold to form the final circuit.

**Edge Component Selection Methods:**

- **Automated Circuit DisCovery (ACDC)** (Conmy et al., 2023): ACDC traverses the transformer's computational graph in reverse topological order, iteratively assigning scores to edges and pruning the circuit.

- **Edge Attribution Patching (EAP)** (Nanda, 2023): EAP is an edge version of AP that takes into account the activations of both the source and target nodes.

We also compared two variants of IBCircuit, namely **IBCircuit-woMI** and **IBCircuit-onlyMI** in **RQ2**, which represent IBCircuit models trained solely with KL loss and MI loss, respectively. Further details about experimental setup are in Appendix B.

**Metrics** For the IOI task, we use *Logit Difference* for evaluation. *Logit Difference* measures the difference in logits assigned to the correct and incorrect answers. For example, for the input "When Mary and John went to the store, John gave a drink to," we calculate logit(Mary)-logit(John). The larger the *Logit Difference*, the better the performance of the model or circuit. In the Greater-Than task, we use the *Greater Probability* metric, which sums the total probability assigned to all correct and incorrect options and calculates the difference, e.g., for the input "The war lasted from the year 1741 to the year 17", we calculate $\sum_{y>41} P(y) - \sum_{y\leq41} P(y)$. A larger difference indicates better model or circuit performance. For these two tasks, we calculate the *KL Divergence* between the logits output by the Circuit and the pre-trained model. A smaller *KL Divergence* indicates that the results

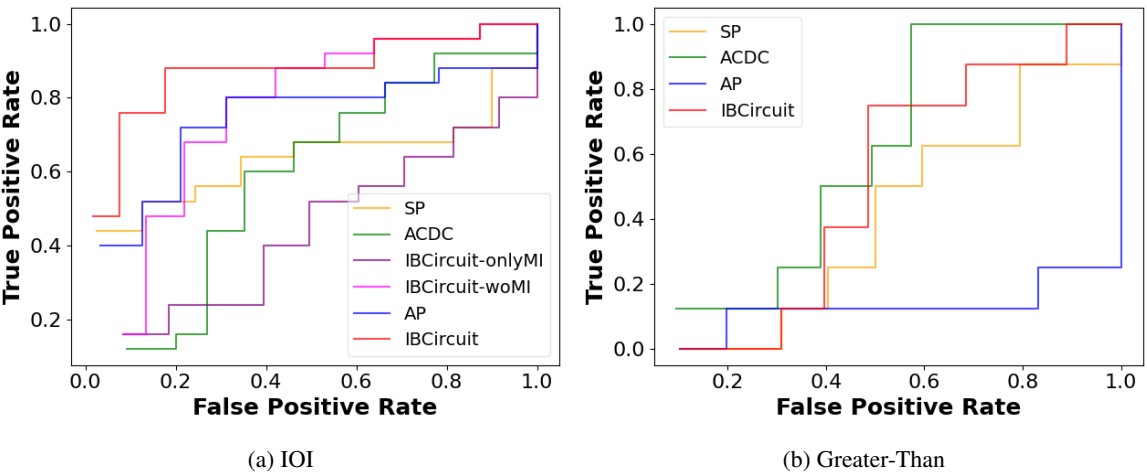

(a) IOI

(b) Greater-Than

*Figure 2.* ROC curves of SP, ACDC, AP and IBCircuit identifying model components from previous work, across IOI circuit and Greater-Than circuit.

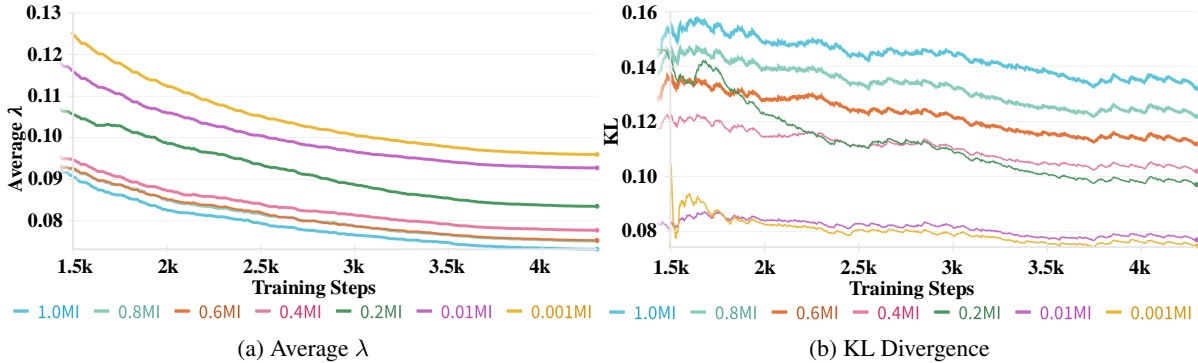

(a) Average $\lambda$

(b) KL Divergence

*Figure 3.* Comparison of the impact of different trade-off coefficients $\alpha$ on IBCircuit in implementing the IOI task.

of the Circuit have greater faithfulness.

**Circuit Ablation** We ablate the nodes that are not included in the circuit by using activation patching to evaluate the effectiveness of identified circuit. We implement randomly selected activations from the corrupted dataset for patching. In IOI task, we construct corrupted inputs by replacing IO and S with arbitrary names. In Greater-Than task, the start year's last two digits are changed to "01", leading models to output years prior to the start year.

### 5.2. RQ1-Grounded in Previous Work

Following (Conmy et al., 2023), we formulate circuit discovery as a binary classification problem, where nodes are classified as positive (in the canonical circuit taken from previous works) or negative (not in the canonical circuit). We determine a series of thresholds for ACDC, SP, AP, and IBCircuit by varying the number of nodes from 10% to 100%, increasing by 10% each time. We plot the pessimistic segments between the Pareto frontiers of TPR and FPR for each method across this range of thresholds.

Figure 2 illustrates the performance of IBCircuit in recovering the canonical circuit within GPT2-small, compared to existing methods. Our findings are as follows: i) IBCircuit shows competitive performance on both the IOI and Greater-Than tasks, notably outperforming baseline methods on the IOI task; ii) however, IBCircuit underperforms compared to ACDC on the Greater-Than task. This discrepancy might be attributed to the fact that the Greater-Than task, unlike the IOI task, does not have a clearly defined expected output. The broader range of possible correct outputs could potentially increase the learning difficulty of the IBCircuit.

### 5.3. RQ2-Ablation Study

**The Influence of KL Loss and MI Loss** In Figure 2, we compare the IBCircuit models trained without KL loss or MI loss. We find that: on the IOI task, IBCircuit outperforms IBCircuit-woMI and significantly surpasses IBCircuit-onlyMI. This can be intuitively explained using Information Bottleneck, as the KL loss primarily serves to align the performance of the noisy model with that of the pretrained

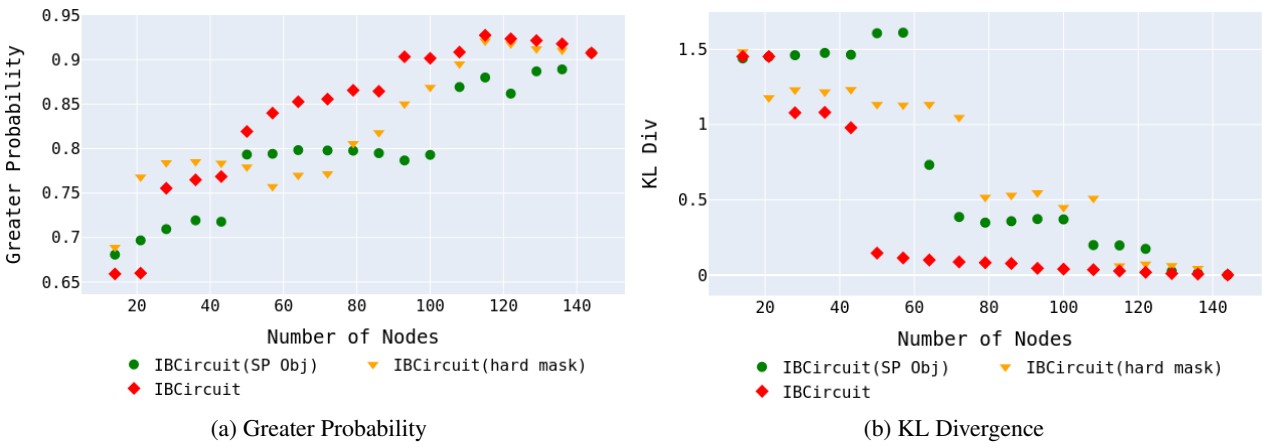

(a) Greater Probability

(b) KL Divergence

*Figure 4.* Comparison of IBCircuit and ablation variants in terms of *Greater Probability* and *KL Divergence* metrics.

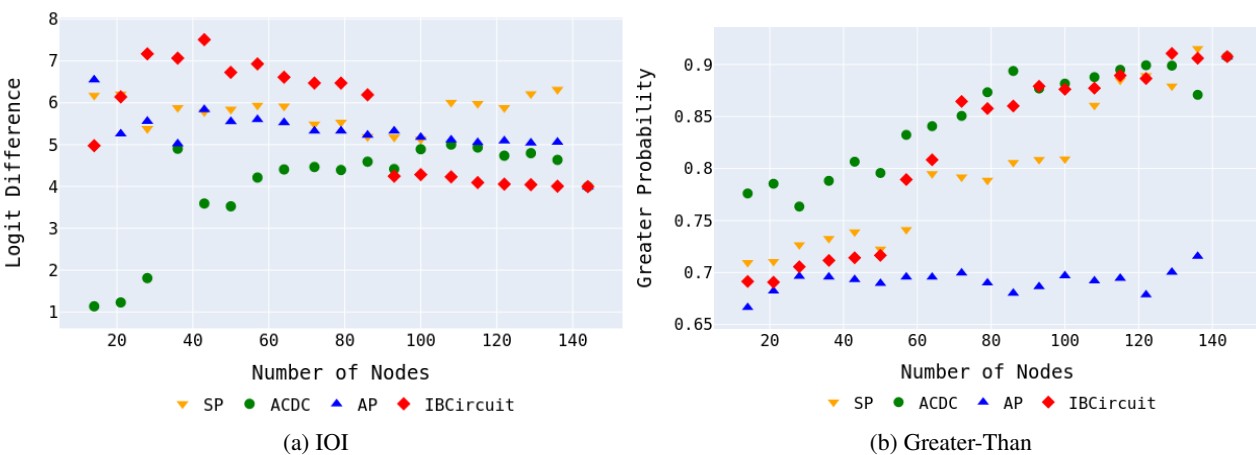

(a) IOI

(b) Greater-Than

*Figure 5.* Comparison of IBCircuit and related methods in terms of *Logit Difference* and *Greater Probability* metrics under different node number thresholds. Higher metric scores and fewer nodes correspond to better circuits.

model, while the MI loss helps reduce irrelevant information in the noisy model. Consequently, the absence of MI loss in IBCircuit-woMI results in slightly worse performance compared to IBCircuit, whereas the lack of KL loss in IBCircuit-onlyMI severely diminishes the performance.

**Parameter Sensitivity Analysis** We further compare the impact of different trade-off coefficients $\alpha$ on IBCircuit in implementing the IOI task in Figure 3. In Figure 3a, we found that as $\alpha$ increases, IBCircuit can achieve a smaller average $\lambda$ with the same number of training steps, indicating that a larger $\alpha$ can learn a sparser Circuit. Correspondingly, as shown in Figure 3b, as $\alpha$ increases, the Circuit obtained by IBCircuit with the same number of training steps has a larger KL Divergence. This indicates that during the learning process, IBCircuit will sacrifice KL divergence to obtain a sparser Circuit. How to set the hyperparameter $\alpha$ depends on whether the user wants to obtain a sparser Circuit or a Circuit with more stable KL divergence.

**Contribution of Each Component** To validate the contribution of each component in IBCircuit, we further compare the following two variants:

- **IBCircuit (hard mask)**: We replaced the *sigmoid-based mask* used for Gaussian perturbations with *Hard-concrete Masking* (Cao et al., 2021).

- **IBCircuit (SP Obj)**: We substituted the mutual information-based loss $L_{MI}$ with the *SP (Subnetwork Probing (Cao et al., 2021)) Objective*, which penalizes the mask based on the probability of it being non-zero.

We conducted two comparative experiments on the **Greaterthan** task using the *Greater Probability* and *KL Divergence* metrics. The evaluation results for these two metrics are shown in Figure 4. Higher *Greater Probability* (Lower *KL Divergence*) and fewer nodes correspond to better circuits.

**Gaussian Perturbations vs. Hard-concrete Masking.** We

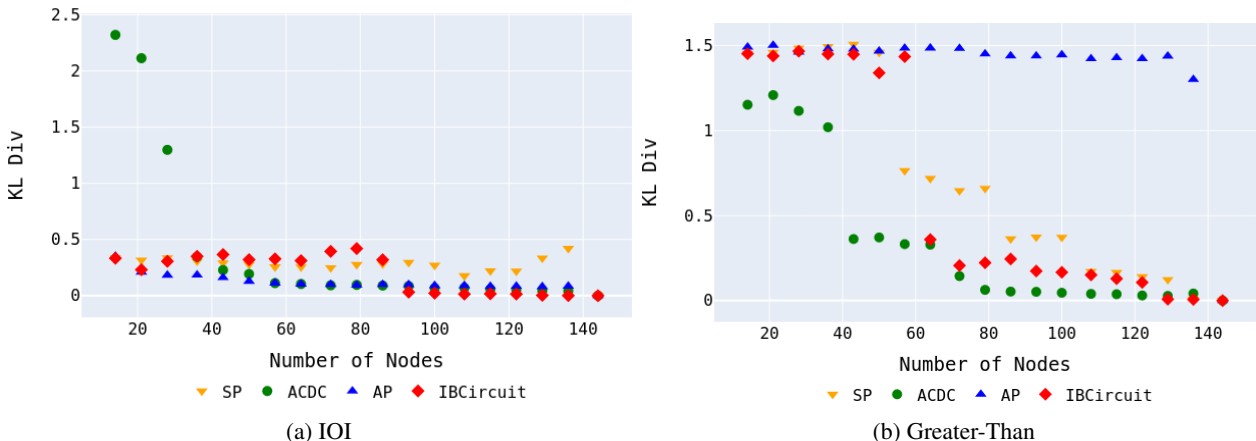

*Figure 6.* Comparison of IBCircuit and related methods in terms of *KL Divergence* metric under different node number thresholds. Lower metric scores and fewer nodes correspond to better circuits.

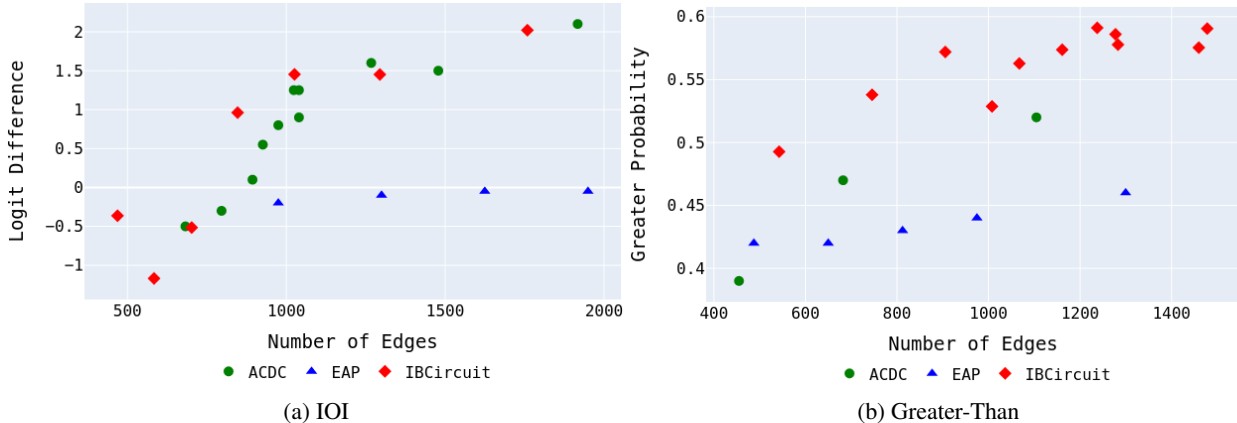

*Figure 7.* Comparison of IBCircuit and related methods in terms of *Logit Difference* and *Greater Probability* metrics under different edge number thresholds. Higher metric scores and fewer edges correspond to better circuits.

observed that, except for cases with a few nodes (fewer than about 45), the circuits optimized via Hard-concrete Masking achieved better *Greater Probability* compared to IBCircuit. However, in such small-node scenarios, all methods resulted in high *KL Divergence* between the identified circuits and the original pretrained model's outputs, indicating significant degradation of the circuit's capabilities. Since preserving the pretrained model's performance is critical, extreme cases with very few nodes are less meaningful. Focusing on scenarios where pretrained capabilities are maintained (cases with more nodes), IBCircuit outperforms the Hard-concrete Masking variant, demonstrating the superiority of Gaussian Perturbations.

**Mutual Information Regularization vs. SP Objective.** The model optimized with the *SP Objective* performed comparably to IBCircuit only when the number of nodes is fewer than 20. In all other cases, it shows significantly worse performance in both *Greater Probability* and *KL Divergence*.

This highlights that the mutual information-based regularization ($L_{MI}$) is more effective at preserving the pretrained model's capabilities while optimizing circuit discovery.

### 5.4. RQ3-Faithfulness & Minimality

Intuitively, a circuit with fewer nodes or edges that still achieves high metrics is less likely to contain components that do not participate in the behavior (Conmy et al., 2023). We measure the performance of various methods in terms of *Logit Difference*, *Greater Probability*, and *KL Divergence* under different node and edge number thresholds.

Figure 5 and Figure 6 present a comparison of metric scores for various methods across different numbers of node components in the Indirect Object Identification (IOI) and Greater-Than tasks. In the IOI task, IBCircuit achieves a higher *Logit Difference* with fewer nodes, demonstrating superior efficiency in utilizing limited components to maintain

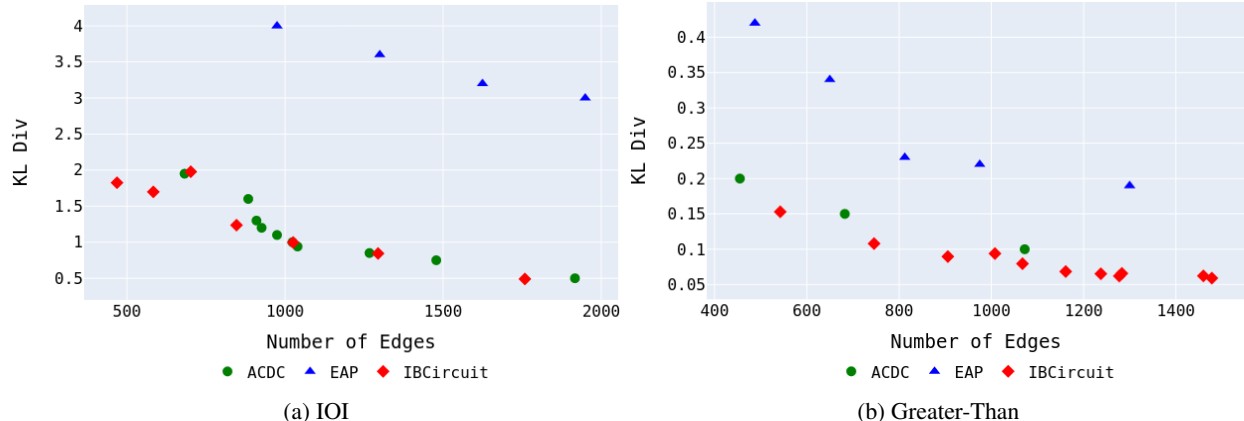

(a) IOI                                      (b) Greater-Than

*Figure 8.* Comparison of IBCircuit and related methods in terms of *KL Divergence* metric under different edge number thresholds. Lower metric scores and fewer edges correspond to better circuits.

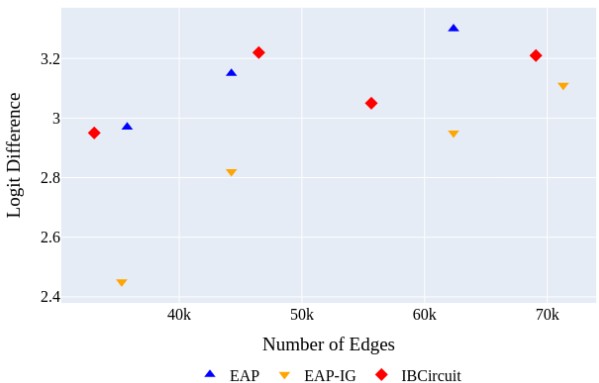

*Figure 9.* Comparison of IBCircuit and related methods in terms of *Logit Difference* metric under different edge number thresholds. Higher metric scores and fewer edges correspond to better circuits.

predictive accuracy. Although IBCircuit does not perform as well as other methods when the number of nodes is greater than 90, it still maintains a large *Logit Difference* (around 4). This may not affect its performance compared to the pretrained model. **Additionally, the proposed method does not require manually designing corrupted data.** In Figure 6, a larger number of nodes achieved a lower *KL Divergence*, which also supports the idea that when the *Logit Difference* is large enough, minor differences do not affect the performance of the Circuit. In the Greater-Than task, when the number of nodes is greater than 70, IBCircuit shows comparable performance to ACDC in terms of *Greater Probability* and *KL Divergence*. However, when the number of nodes is low, IBCircuit performs worse than ACDC. This may be because the Greater Than task does not have a specific token as the answer, unlike the IOI task. When the model's sparsity is highly pursued, IBCircuit finds it difficult to achieve the best faithfulness.

For edge component selection presented in Figure 7 and Figure 8, IBCircuit outperforms baseline methods in both IOI and Greater-Than tasks. In the IOI task, as the number of edges decreases, IBCircuit consistently achieves a higher *Logit Difference* and lower *KL Divergence* compared to ACDC and EAP. Notably, in the Greater-Than task, IBCircuit maintains a *Greater Probability* above 50% and a *KL Divergence* below 0.15 even with very sparse edge counts. This highlights its efficiency and reliability in optimizing circuit performance under constrained components.

### 5.5. RQ4-Scalability to Large Models

Following the experimental setup of EAP-IG (Hanna et al., 2022), we compare IBCircuit with EAP-IG and EAP on the IOI task using GPT-2 XL (1.5B) (Radford et al., 2019). As shown in the Figure 9, IBCircuit achieves performance comparable to EAP and a better *logit difference* than EAP when the number of edges ranges from 30k to 70k (approximately 97%-98.5% of the total 2235025 edges). This result demonstrates the scalability of IBCircuit to larger models.

## 6. Conclusion

In this paper, we aim to address the challenge of understanding the behavior of Transformer-based models, which are often seen as black boxes due to their complex computations. In this work, we propose an end-to-end approach based on the principle of Information Bottleneck, called IBCircuit, to identify informative circuits holistically. IBCircuit is an optimization framework for holistic circuit discovery and can be applied to any given task without tediously corrupted activation design. In both the Indirect Object Identification (IOI) and Greater-Than tasks, IBCircuit identifies more faithful and minimal circuits in terms of critical node components and edge components.

## Acknowledgment

This research is supported by grants from the Research Grants Council of the Hong Kong Special Administrative Region, China (No. CUHK 14217622). This work was supported by Damo Academy (Hupan Laboratory) through Damo Academy (Hupan Laboratory) Innovative Research Program. The authors would like to express their gratitude to the reviewers for their feedback, which has improved the clarity and contribution of the paper.

**Equal Contribution Statement** Tian Bian and Yifan Niu jointly proposed the IBCircuit framework. Yifan Niu introduced variational approximation, derived the theoretical results, and wrote Section 1-4. Both of them contributed to the early development of the IBCircuit code. Tian Bian then extended the code to the IOI and greater-than tasks, conducted the experiments on GPT-2, and wrote Section 5. Both authors contributed equally to this work; listed alphabetically by last name.

## Impact Statement

This paper presents work whose goal is to advance the field of mechanistic interpretability. There are many potential societal consequences of our work, none of which we feel must be specifically highlighted here.

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

# A. Analysis of IBCircuit

## A.1. Proof of IBCircuit Objective

We justify the above formulation through the following derivation. Let $\mathcal{G}_s$ be a subset of $\mathcal{G}$, which is independent to $Y$. Denote $\mathcal{G}_\epsilon$ as the noisy subset of $\mathcal{G}$ determined by injected noise, if we select the circuit $\mathcal{C}$ by dropping $\mathcal{G}_\epsilon$ in $\mathcal{G}$, the following inequality holds:

$$I(\mathcal{G}_s; \mathcal{C}) \leq I(\mathcal{G}_s; \hat{\mathcal{G}}) \leq I(\mathcal{G}; \hat{\mathcal{G}}) - I(Y; \hat{\mathcal{G}}). \tag{14}$$

This equation indicates when setting $\alpha = 1$ in Eq. (8), the IBCircuit objective upper bounds the mutual information of $\mathcal{G}_s$ and $\mathcal{C}$. Hence, optimizing the IBCircuit objective encourages $\mathcal{C}$ to be less related to components in $\mathcal{G}_s$ which are irrelevant to $Y$.

*Proof.* We follow the proof in (Yu et al., 2022). Suppose $\mathcal{G}$, $\mathcal{C}$, $\mathcal{G}_s$ and $Y$ satisfy the Markov condition $(Y, \mathcal{G}_s) \to \mathcal{G} \to \mathcal{C}$ (Achille & Soatto, 2018a). Then we have the following inequality:

$$I(\mathcal{C}; \mathcal{G}) \geq I(\mathcal{C}; Y, \mathcal{G}_s) = I(\mathcal{C}; \mathcal{G}_s) + I(\mathcal{C}; Y | \mathcal{G}_s). \tag{15}$$

Since $Y$ and $\mathcal{G}_s$ are independent, we have $H(Y | \mathcal{G}_s) = H(Y)$ and $H(Y | \mathcal{G}_s, \mathcal{C}) \leq H(Y | \mathcal{C})$. Then we have:

$$I(\mathcal{C}; Y | \mathcal{G}_s) = H(Y | \mathcal{G}_s) - H(Y | \mathcal{G}_s, \mathcal{C}) \geq H(Y) - H(Y | \mathcal{C}) = I(\mathcal{C}; Y) \tag{16}$$

Combine Eq. (15) and Eq. (16), we:

$$I(\mathcal{C}; \mathcal{G}_s) \leq I(\mathcal{C}; \mathcal{G}) - I(\mathcal{C}; Y) \tag{17}$$

Suppose $\mathcal{G}$, $\hat{\mathcal{G}}$, $\mathcal{G}_s$ and $Y$ satisfy the Markov condition $(Y, \mathcal{G}_s) \to \mathcal{G} \to \hat{\mathcal{G}}$ (Achille & Soatto, 2018a). Then, combine with Eq. (17) we have:

$$I(\hat{\mathcal{G}}; \mathcal{G}_s) \leq I(\mathcal{G}_s; \mathcal{G}) - I(\hat{\mathcal{G}}; Y) \tag{18}$$

$\hat{\mathcal{G}}$ is deterministic given $\mathcal{G}_\epsilon$ and $\mathcal{C}$, since we can recover $\hat{\mathcal{G}}$ by combining $\mathcal{G}_\epsilon$ with $\mathcal{C}$. Then for the left part in Eq. (18), we have:

$$I(\hat{\mathcal{G}}; \mathcal{G}_s) = I(\mathcal{G}_\epsilon, \mathcal{C}; \mathcal{G}_s) = I(\mathcal{C}; \mathcal{G}s) + I(\mathcal{C}; \mathcal{G}_\epsilon | \mathcal{G}_s) \geq I(\mathcal{C}; \mathcal{G}_s) \tag{19}$$

Therefore, by combining Eq. (18) and Eq. (19) we have the follow inequality:

$$I(\mathcal{C}; \mathcal{G}_s) \leq I(\hat{\mathcal{G}}; \mathcal{G}_s) \leq I(\hat{\mathcal{G}}; \mathcal{G}) - I(\hat{\mathcal{G}}; Y) \tag{20}$$

which proofs Eq. (14). $\square$

## A.2. Derivation of MI Loss

The Kullback-Leibler (KL) divergence is a measure of how one probability distribution diverges from a second, expected probability distribution. In this section, we derive the KL divergence formula between two Gaussian distributions $Q_i$ and $P_i$, and further calculate the average KL divergence over a batch of data.

### A.2.1. KL DIVERGENCE FORMULA

Given two Gaussian distributions: $P_i : \mathcal{N}(\mu_i, \sigma_i)$ and $Q_i : \mathcal{N}(\lambda_i h_i + (1 - \lambda_i)\mu_i, (1 - \lambda_i)\sigma_i)$. The KL divergence $D_{\mathrm{KL}}(Q_i \| P_i)$ is given by:

$$D_{\mathrm{KL}}(Q_i \| P_i) = \frac{1}{2} \left( \log \frac{\sigma_{P,i}^2}{\sigma_{Q,i}^2} + \frac{\sigma_{Q,i}^2 + (\mu_{Q,i} - \mu_{P,i})^2}{\sigma_{P,i}^2} - 1 \right) \tag{21}$$

Substituting the parameters:

$$D_{\mathrm{KL}}(Q_i \| P_i) = \frac{1}{2} \left( \log \frac{\sigma_i^2}{(1 - \lambda_i)^2 \sigma_i^2} + \frac{(1 - \lambda_i)^2 \sigma_i^2 + (\lambda_i h_i + (1 - \lambda_i)\mu_i - \mu_i)^2}{\sigma_i^2} - 1 \right) \tag{22}$$

Simplifying the terms:

$$D_{\text{KL}}(Q_i\|P_i) = \frac{1}{2}\left(\log\frac{1}{(1-\lambda_i)^2} + \frac{(1-\lambda_i)^2\sigma_i^2 + \lambda_i^2(h_i-\mu_i)^2}{\sigma_i^2} - 1\right) \tag{23}$$

$$= -\log(1-\lambda_i) + \frac{(1-\lambda_i)^2 - 1}{2} + \frac{\lambda_i^2(h_i-\mu_i)^2}{2\sigma_i^2} \tag{24}$$

### A.2.2. AVERAGE KL DIVERGENCE OVER A BATCH

For a batch of $n$ data points, the average KL divergence is:

$$D_{\text{KL}} = \frac{1}{n}\sum_{i=1}^{n} D_{\text{KL}}(Q_i\|P_i) \tag{25}$$

Substituting the expression for $D_{\text{KL}}(Q_i\|P_i)$:

$$D_{\text{KL}} = \frac{1}{n}\sum_{i=1}^{n}\left(-\log(1-\lambda_i) + \frac{(1-\lambda_i)^2 - 1}{2} + \frac{\lambda_i^2(h_i-\mu_i)^2}{2\sigma_i^2}\right) \tag{26}$$

Breaking down the summation:

$$D_{\text{KL}} = -\sum_{i=1}^{n}\frac{1}{n}\log(1-\lambda_i) + \sum_{i=1}^{n}\frac{(1-\lambda_i)^2 - 1}{2n} + \sum_{i=1}^{n}\frac{\lambda_i^2(h_i-\mu_i)^2}{2n\sigma_i^2} \tag{27}$$

Assuming $A_{\mathcal{G}} = -\sum_{i=1}^{n}(1-\lambda_i), B_{\mathcal{G}} = \sum_{i=1}^{n}\frac{\lambda_i(h_i-\mu_i)}{\sigma_i}$:

$$D_{\text{KL}} = -\frac{1}{n}\log A_{\mathcal{G}} + \frac{(A_{\mathcal{G}})^2 - 1}{2n} + \frac{B_{\mathcal{G}}^2}{2n} \tag{28}$$

## B. Experimental Setting

For node-level circuit discovery, we set the learning rate to 0.05, trained for 1300 epochs using the Adam optimizer for IB weights, and set $\alpha$ to 1. For edge-level circuit discovery, we set the learning rate to 0.1, trained for 3000 epochs using the Adam optimizer for IB weights with a learning rate warm-up scheduler (200 warm-up steps), and set $\alpha$ between 0.01 and 1 to balance the sparsity of the circuit and the KL divergence (see the analysis of the parameter $\alpha$ in Section 5.3). To obtain the performance of IBCircuit with fewer edges, we selected higher values of $\alpha$.

