# OpenReview forum: "IBCircuit: Towards Holistic Circuit Discovery with Information Bottleneck"
_ICML.cc/2025/Conference — ICML 2025 poster_

### Official Review · Reviewer_ZTi2 · 2025-03-05

**Overall Recommendation:** 3

**Summary:**

This paper proposes a method, IBCircuit, for discovering circuits based in the information bottleneck method. Specifically, by interpolating between the original activation and the mean batch activation of a component or edge, and learning these coefficients for all nodes or edges simultaneously, one can discover circuits without needing to design counterfactual activations manually (as is often done in prior work).

The proposed optimizes two objectives: one that encourages the circuit $\mathcal{C}$ to be informative on output $Y$, and one that encourages the circuit $\mathcal{C}$ to be sparse. It is found that the proposed method achieves better AUROC than prior methods, that the sparsity term is slightly helpful (but not useful on its own), and that the proposed method is generally better for preserving the full model’s task-specific information—more so on IOI than Greater-Than.

**Claims And Evidence:**

The main claims are supported by a reasonable amount of quantitative evidence. My main concern is that the circuits found in the IOI and Greater-Than papers are taken as ground-truth circuits; see Methods and Evaluation.

It is also claimed that one of the main contributions is that we do not need to manually construct task-specific counterfactual activations. But this is already common: mean ablations, which are used in many circuit discovery papers, do not require constructing targeted counterfactual dataset pairs, but instead simply taking the mean activation of a component over some general text corpus. One can instead define the mean as over a task dataset, but this does not have to be the case—it simply performs better this way. Optimal Ablations [4; references provided under “Essential References” section] also does not require task-specific activations. With respect to the “manually constructed” part, I’m not sure what this means, exactly: patching with an activation from a counterfactual input does not require us to manually design the activation either; we just need to provide a counterfactual input and the model will produce the new activation automatically. Also, in the evaluation (Sec. 5), don’t we need counterfactual activations from other inputs to evaluate the circuits?

**Essential References Not Discussed:**

* Activation patching: cite Vig et al. [1], Finlayson et al. [2].
* When referring to adding Gaussian noise to a component, cite Meng. They are already cited elsewhere in the paper, but this method is not attributed to them as it should be.
* Cite information flow routes [3]. This method should be discussed and contrasted with the proposed method.
* Cite optimal ablations [4] when referring to ablating multiple components at once and learning coefficients on them. The UGS method they propose also involves interpolating between real activations and noise; as far as I can tell, the main difference between UGS and this method is that UGS involves learning the ablation value, rather than defining it as the mean within a batch. Otherwise, they are very similar.

References:

[1] Vig et al. (2020). “Investigating Gender Bias in Language Models Using Causal Mediation Analysis.” NeurIPS. https://proceedings.neurips.cc/paper/2020/hash/92650b2e92217715fe312e6fa7b90d82-Abstract.html

[2] Finlayson et al. (2021). “Causal Analysis of Syntactic Agreement Mechanisms in Neural Language Models.” ACL. https://aclanthology.org/2021.acl-long.144/

[3] Ferrando & Voita (2024). “Information Flow Routes: Automatically Interpreting Language Models at Scale.” arXiv (posted ~1 year before ICML submission deadline, then published at EMNLP within the 4-month  window). https://aclanthology.org/2024.emnlp-main.965/

[4] Li & Janson (2024). “Optimal ablation for interpretability.” arXiv (posted > 4 months before ICML submission deadline, then published at NeurIPS within the 4-month window). https://openreview.net/forum?id=opt72TYzwZ

**Experimental Designs Or Analyses:**

Evaluating circuits with counterfactual activations from alternate inputs is not very similar to the way that the circuit is discovered. This could unfairly penalize methods that discover circuits using very different kinds of activation patching, including the baselines. This actually strengthens the argument in favor of IBCircuits, as being able to generalize to new kinds of patching methods suggests that the discovered circuit is more robust. Nonetheless, I think this should be discussed explicitly, as it doesn’t seem like a fair comparison across all methods; the performance will probably correlate with the similarity of the discovery and evaluation patching methods.

**Methods And Evaluation Criteria:**

In Figure 2, the circuits of Wang et al. for IOI and Hanna et al. for Greater-Than are treated as ground-truth circuits. These two circuits may be *high-precision* circuits that were manually validated, they are not guaranteed (nor likely, in my opinion) to have recalled all of the important causal dependencies. If this is true, then certain methods maybe unfairly penalized for recovering components that actually were important, but were simply not found by Wang et al. (2023) nor Hanna et al. (2023).

The KL Divergence is also a very broad metric that recovers far more than just task-specific behavior. A sparser circuit will generally have lower faithfulness, even if it perfectly recovers task performance. The logit difference is a good metric, so I don’t know whether targeting low KL divergence will be helpful in addition to the other metrics proposed.

**Other Comments Or Suggestions:**

* L78: rephrase final clause of the sentence at end of Sec. 2? I wasn’t sure what this meant.
* Could T be added as a subscript to Y in Sec. 4 to make it clear which tasks Y is over?
* Use \mathcal for C and G in the appendix for consistency.

**Other Strengths And Weaknesses:**

Strengths:
* I like that the performance of the circuit and the faithfulness of the circuit to the full model are separate metrics. This should be more common.
* The writing is generally easy to follow.

Weaknesses:
* Figure 1 is not very clear. Maybe color could be used in a more consistent way across both nodes and edges to indicate noising? And maybe the default vs. noising colors could be made more distinct from each other?

**Questions For Authors:**

* Appendix B: This is quite a few epochs. What was the runtime of your method, and on what kind of GPU? I just want to verify that it’s at least an order-of-magnitude better runtime than ACDC (which is an easy bar to pass), and verify scalability.

**Relation To Broader Scientific Literature:**

The mechanistic interpretability essentials are discussed and well contextualized. However, there is a line of highly related recent work (over 4 months from the submission deadline) that is not cited, but should be. Perhaps most related is the Optimal Ablations paper [4], which proposes a method called UGS that functions similarly to the coefficient learning approach here. Information Flow Routes [3] is conceptually similar in that it uses information-theoretic notions to define a circuit discovery method, but the actual method is quite different.

Additionally, the use of noise to ablate components was proposed by Meng et al. (who are cited in the paper), but this is not directly attributed to them and should be. Also cite the original activation patching papers: Vig et al. [1] and Finlayson et al. [3].

See Essential References for references.

**Theoretical Claims:**

Proofs in appendix look good to me.

---

> ### Author Rebuttal · Authors · 2025-04-01
>
> We sincerely appreciate your time and effort in reviewing our paper and providing constructive feedback. We would like to address your questions and concerns below.
> >**Claims and Evidence**: **Q1**: Not sure what claims of avoiding manually constructed counterfactual activations mean. **Q2**: In the evaluation, don’t we need counterfactual activations from other inputs to evaluate the circuits?
>
> **Q1**: We emphasize that our circuit discovery process does not require manually designing counterfactual inputs. While mean ablation also avoids explicit counterfactual input design, it underperforms the resampling ablation [1]. Additionally, we agree with the comment that "the performance will probably correlate with the similarity of the discovery and evaluation patching methods"[2]. The correlation arises from the biases introduced by different counterfactual inputs as corrupted data. In contrast, IBCircuit leverages Gaussian as corrupted activations rather than corrupted data, which eliminates such biases.
>
> **Q2**: To ensure fair comparison, we follow existing baselines and utilize counterfactual activations derived from other inputs during evaluation. This demonstrates that our method achieves robust results even when assessed using ablation strategies distinct from the ablation employed in circuit discovery.
>
> [1] Causal Scrubbing: a method for rigorously testing interpretability hypotheses.
>
> [2] Transformer Circuit Faithfulness Metrics Are Not Robust. COLM2024.
>
>
> >**Methods And Evaluation Criteria**: **Q1**:Treating prior circuits as "ground-truth" risks missing valid components. **Q2**: KL divergence recovers far more than just task-specific behavior.
>
> **Q1**: We agree with your opinion that treating manually validated circuits as ground truth may disadvantage newer discoveries. However, since baselines [1][2] also evaluate against these manually validated results as ground truth, we followed their protocol to ensure fair comparison. Beyond this Grounded in Previous Work experiment, we further demonstrate the advantages of our method through comparative analyses on metrics such as Faithfulness and Minimality.
>
> **Q2**: In [1], Appendix C ("Discussion of Metrics Optimized") argues that:(1) Optimizing for low KL divergence is the simplest and most robust metric applicable across diverse tasks;(2) KL divergence is universally applicable to next-token prediction tasks because it does not require predefined output labels (unlike metrics such as logit difference, which necessitate specifying the exact tokens between which differences are computed). Based on these insights, we adopted KL divergence as our optimization metric.
>
> [1] Towards Automated Circuit Discovery for Mechanistic Interpretability. NeurIPS 2023.
>
> [2] Attribution Patching Outperforms Automated Circuit Discovery. NeurIPS 2023.
>
> >**Experimental Designs Or Analyses**: Evaluation uses counterfactual activations dissimilar to baseline discovery methods.
>
> **We would like to clarify that our evaluation is the same as the existing methods.** We agree with the statement that "the performance will probably correlate with the similarity of the discovery and evaluation patching methods". For this reason, in our experiments, we adopt unified counterfactual activations derived from alternate inputs for all baseline methods, as well as our proposed approach to ensure fair comparison. The proposed method is robust and does not rely on this ablation strategy for circuit discovery. We appreciate your suggestion and will explicitly discuss this point in the revision.
>
>
> >**Citation and writing issues mentioned in the weakness and suggestions.**
>
> Thank you for your suggestion. We will revise the corresponding part in the new version. We will discuss and cite the recommended papers.
>
>
> >**Question**: Runtime and GPU.
>
> The IBCircuit runs the IOI task on one H20 GPU for about 1.2 hours. The official implementation of ACDC takes about 16 hours on the same device. We will add the comparison in the revised version.

---

> > ### Comment · Reviewer_ZTi2 · 2025-04-02
> >
> > Thanks for the detailed response!
> >
> > > **Claims and Evidence**
> >
> > Q1: Got it, so the claim is more that we don't need manually designed counterfactuals, *and* it's on par with/better than when we do have them. This sounds good to me.
> >
> > Q2: Following precedent is fine if it does not interfere with the evidence for the claims; here, I agree that it emphasizes how well your circuits generalize. That said (and this is admittedly a nitpick), I'm not sure this is necessarily the most fair comparison. One could also evaluate circuits by adding noise to components outside the circuit; this would more closely resemble your circuit discovery setting. One could also evaluate by setting them to their mean values, or by learning ablation values that preserve low loss. It seems like an arbitrary choice.
> >
> > > **Methods and Evaluation Criteria**
> >
> > Q1: While past work has done this, I also think their evaluations are flawed for it. If we really want to ensure that our circuits are good, we should use a model that contains a known ground-truth circuit. For examples of what this could look like, see InterpBench [1].
> >
> > Q2: KL divergence is universally applicable for the same reason that it's not precise. Cross-entropy given is also universally applicable, and its values are more specific to relevant tokens instead of all of them. I reiterate that in circuit discovery, we are trying to isolate task-specific circuits, and that we should therefore not mind if the rest of the distribution is heavily affected by the ablations.
> >
> > > **Experimental Designs or Analysis**
> >
> > Precedent is not always a good reason. New methods can sometimes require new methods to ensure fair comparisons. As above, I think the current evidence already supports the value of your method; I just don't think that it's necessarily the most fair comparison. I would be happy with a discussion of the advantages and disadvantages of your choice of evaluation technique, just so readers have a fair discussion for reference.
> >
> > References:
> >
> > [1] Gupta et al. (2024). "InterpBench: Semi-synthetic Transformers for Evaluating Mechanistic Interpretability Techniques." NeurIPS Datasets and Benchmarks.

---

> > > ### Author Response · Authors · 2025-04-09
> > >
> > > Thank you for raising this critical point. We fully agree that exploring robust and fair evaluation methodologies remains an open challenge in the circuit analysis community. Recent works such as [1] (we supplement the comparison with the EAP-IG proposed in [1] in the [anonymous link](https://anonymous.4open.science/r/IBCircuit_ICML25_rebuttal-76E3/Experimental%20Results%20for%20Reviewer%20tPrr/More%20tasks,%20scalability%20to%20Larger%20Models,%20and%20comparison%20with%20EAP-IG.md)) and [2] further highlight the fragility of existing metrics and emphasize the need for standardized and nuanced evaluation frameworks. While our current experiments align with established baselines for reproducibility, we acknowledge the limitations of existing ablation evaluation. As a complementary approach, downstream application tests such as leveraging circuit insights for parameter-efficient fine-tuning[3] are worth exploring. We will incorporate InterpBench and explore these application-driven evaluations into our future work to rigorously validate circuit faithfulness. Your suggestions greatly enhance the depth of our analysis, and we sincerely appreciate your constructive feedback.
> > >
> > > [1] Have Faith in Faithfulness: Going Beyond Circuit Overlap When Finding Model Mechanisms, COLM2024
> > >
> > > [2] Transformer Circuit Faithfulness Metrics are not Robust, COLM2024
> > >
> > > [3] ReFT: Representation Finetuning for Language Models, NeurIPS2024

---

### Official Review · Reviewer_vc6F · 2025-03-10

**Overall Recommendation:** 3

**Summary:**

This paper proposes a method to identify circuits -- interpretable subgraph of a computation graph of a neural net -- via the information bottleneck principle. The key idea of the paper's' technique is estimating the IBCircuit objective via noise injection. Empirical results on several well-known tasks, such as IOI and Greater-Than, demonstrate that the method can faithfully (measured by e.g. the logit diff metric) recover circuits with fewer circuit components compared to previous methods.

**Claims And Evidence:**

I find that the circuit finding performance (measured in recovery metric and circuit components) is generally well supported with the paper's experiments. The authors have shown that IBCircuit can be (resp.) relative or outperform well-established techniques (e.g. ACDC) in terms of false positive rates on GT and IOI tasks. In the low-node regime, IBCircuit can also outperform these techniques in terms of recovery metrics.

However, I do find that the completely foregoing corrupted data to be a bit troubling. The key idea of such a contrast set is to causally intervene on the model's behavior, so that the method can identify model components that are meaningfully contributing to the actual implementation of an algorithmic task. Here, I'd argue that the author is identifying a slightly different subgraph (which may or may not be the circuit), which may or may not be causal to the implementation of this task, and I'd encourage the author to discuss more on these subtleties.

There is also a small error in the author's claim that the authors of IOI [1] and Greater-Than [2] have used mean dataset values for their circuit discovery analyses (line 93), where they have instead used permutation sampling to obtain such values. I believe these are subtle but critical points in the design philosophy of the goal of circuit finding. This type of resampling ablation has long been advocated in [3] which precedes both [1][2].

[1] Interpretability in the Wild: a Circuit for Indirect Object Identification in GPT-2 small, https://arxiv.org/abs/2211.00593

[2] How does GPT-2 compute greater-than?: Interpreting mathematical abilities in a pre-trained language model, https://arxiv.org/abs/2305.00586

[3] Causal Scrubbing: a method for rigorously testing interpretability hypotheses, https://www.lesswrong.com/posts/JvZhhzycHu2Yd57RN/causal-scrubbing-a-method-for-rigorously-testing

**Essential References Not Discussed:**

See previous section on **Relation To Broader Scientific Literature**. The authors should reference [1] as it is also a scalable approach, and [2] is a method that the authors cite, but does not properly discuss its independence from contrast set.


[1] Finding Transformer Circuits with Edge Pruning, https://arxiv.org/abs/2406.16778

[2] Sparse Feature Circuits: Discovering and Editing Interpretable Causal Graphs in Language Models, https://arxiv.org/abs/2403.19647

**Experimental Designs Or Analyses:**

Please check my review in the __Methods And Evaluation Criteria__ section, where I discussed some concerns regarding the evaluation metrics.

**Methods And Evaluation Criteria:**

The method is solid. The authors have clearly defined a learning objective and designed feasible implementations.

I believe the paper could benefit from a more rigorous set of evaluations. For example, several works [1][2] have discussed that some circuit evaluation metrics discussed (and used) in this paper are not robust. Given the closeness of some of these results, it'd be good to have statistical tests, or bring in some of the evaluation methods proposed in these papers, for more rigorous testing.

Additionally, the authors have stated that "(existing methods) do not scale well with the model size", but the authors have conducted their experiments on a similar scale (i.e. GPT-2 small). If this is indeed a problem that the authors are aiming to address, they should experiment with larger architectures (e.g. 7B or 13B models) as other scalable circuit finding methods (e.g. edge pruning) have demonstrated [3].

[1] Have Faith in Faithfulness: Going Beyond Circuit Overlap When Finding Model Mechanisms, https://arxiv.org/abs/2403.17806

[2] Transformer Circuit Faithfulness Metrics are not Robust, https://arxiv.org/abs/2407.08734

[3] Finding Transformer Circuits with Edge Pruning, https://arxiv.org/abs/2406.16778

**Other Comments Or Suggestions:**

The writing is generally in good condition.

**Other Strengths And Weaknesses:**

- Strength: clearly grounded methods with information theory support.
- Weakness: lack of proper discussion with prior works (see above), the core philosophy of the goal of circuit finding (can we claim that the circuit found by IBCircuit is causal?), and perhaps more rigor in evaluation.

**Questions For Authors:**

I'm happy to read your take on several points raised above.

**Relation To Broader Scientific Literature:**

The authors have generally situated this paper in classical circuit finding literatures well. But I believe there should be a more direct, head-to-head comparison against some recent developments in circuit finding techniques that are more scalable, e.g. edge pruning [1] and sparse feature circuits [2]. In particular, [1] is able to scale their circuit finding technique to 13B models and [2] does not require a contrast set. There are also some minor errors in the literature review discussed in the __Claims and Evidence__ section.

[1] Finding Transformer Circuits with Edge Pruning, https://arxiv.org/abs/2406.16778

[2] Sparse Feature Circuits: Discovering and Editing Interpretable Causal Graphs in Language Models, https://arxiv.org/abs/2403.19647

**Theoretical Claims:**

IBCircuit is grounded in information theory, which is a good idea to start with. I have checked the authors mathematical claims (e.g. derivations, optimization objectives) and do not see any error.

---

> ### Author Rebuttal · Authors · 2025-04-01
>
> We sincerely appreciate your time and effort in reviewing our paper and providing constructive feedback. We would like to address your questions and concerns below.
> >**Claims and Evidence**: **Q1**. Discuss the causality of circuit identification. **Q2**. Sampling methods in previous circuit studies IOI, Greater-than.
>
> **Q1**: We appreciate your insight on causality. **We would like to clarify that IBCircuit is causal.** As in causal inference literature [1], our method employs controlled noise injection with learned IB weights as implicit causal interventions. And we do not need explicit data corruption. Unlike conventional ablation-based approaches, IBCircuit avoids distributional biases from discovery-phase ablations. Therefore, we can overcome the robustness limitations mentioned in [2] and find more reliable circuits.
>
> **Q2**: Thank you for highlighting this detail. We acknowledge that IOI introduces mean ablation in Section 2.2, while Greater Than uses permutation sampling. We will clarify these distinctions in the revised version.
>
> [1] A Causality-Aware Perspective on Domain Generalization via Domain Intervention.
>
> [2] Transformer Circuit Faithfulness Metrics Are Not Robust. COLM2024.
>
> >**Methods and Evaluation**: **Q1**. Conduct more rigorous statistical tests. **Q2**. Experiment with larger model architectures.
>
> **Q1**:  We provide additional results under various ablation methods in [this anonymous link](https://anonymous.4open.science/r/IBCircuit_ICML25_rebuttal-76E3/), supplementing the random ablation presented in our initial draft. We also include the suggested evaluations.
>
> **Q2**: We add comparison on CodeLLaMA-13B in the same anonymous link above.
>
> >**Broader Scientific Literature and Essential References**: Reference and comparisons with: (1)Edge pruning, (2)Sparse feature circuit.
>
> Thank you for the valuable feedback. We include the comparison with the two baselines in [this anonymous link](https://anonymous.4open.science/r/IBCircuit_ICML25_rebuttal-76E3/).  Due to our limited GPU resources, we have not yet completed all experiments. We will keep the remaining results updated in the next few days. Additionally, we will cite the two papers in the revision.

---

> > ### Comment · Reviewer_vc6F · 2025-04-04
> >
> > Thank you for these additional results. I believe including them would bolster the claims made in this paper. I have adjusted my score to 3 to reflect these changes. I'd urge the authors to include feedback from other reviewers, in particular that of tPrr, to improve the overall presentation and clarity.

---

> > > ### Author Response · Authors · 2025-04-09
> > >
> > > Thank you for your constructive feedback and for acknowledging the value of our additional results. We sincerely appreciate your adjusted score and will carefully incorporate all reviewer suggestions, especially Reviewer tPrr's recommendations on improving presentation clarity, into the revised manuscript. Your input is invaluable in strengthening this work, and we are committed to addressing these points thoroughly in the final version.

---

### Official Review · Reviewer_tPrr · 2025-03-14

**Overall Recommendation:** 3

**Summary:**

This paper addresses the recent surge of interest in discovering circuits inside language models that are sufficient for faithfully explaining the behavior on specific tasks. The key contribution is to provide a method grounded in the Information Bottleneck (IB) method, which is optimized directly (avoiding expensive iterative patching), works without manually constructed corrupted inputs, and outperforms various existing methods.

## update after rebuttal
The authors have clarified key aspects and provided new results. The point remains that the clarity of the paper needs to be very strongly improved with regard to how the variational IB is implemented -- I have increased my score trusting the authors that they will keep their promise to revise this aspect thoroughly.

**Claims And Evidence:**

I believe the claims as made in the Abstract, the Introduction, and the Conclusion are largely supported by evidence.

Regarding the claim that the method outperforms "recent related work" (made in the abstract), I believe comparison to IG-EAG [1] is missing. I discuss this further under "Other Strengths and Weaknesses".

[1] Hanna et al, Have Faith in Faithfulness: Going Beyond Circuit Overlap When Finding Model Mechanisms, COLM 2024

**Essential References Not Discussed:**

All essential references are cited, though I would like to ask the authors to provide further comparison as detailed at other parts of this review.

**Experimental Designs Or Analyses:**

The soundness of the experimental design appears good. My main worry is that the basic design of the method isn't sufficiently clear from the paper (see under Theoretical Claims -- though from the rebuttal it does make sense to me), but beyond this the design seems fine.

**Methods And Evaluation Criteria:**

Yes, the evaluation criteria make sense. The paper uses appropriate standard benchmarks.

A weakness is that evaluation is limited to two tasks (IOI and Greater-Than) and a small model (GPT2-Small), even though a few more relevant tasks are available. In particular, the paper [1] cited above uses six such tasks.

**Other Comments Or Suggestions:**

None

**Other Strengths And Weaknesses:**

Strengths:

- Grounding in IB in principle permits a principled motivation of the approach (though see doubts below under “Theoretical Claims”)


- The idea of a circuit discovery method based on end-to-end optimization is very appealing, and may hold the promise to be much more scalable than  iterative methods such as ACDC.

- The approach is competitive or even better than existing methods compared to.


- It is remarkable that the proposed method performs well even without using explicit corrupted data (one may wonder if it could perhaps work even better if one replaced the Gaussian perturbations with activations computed on randomly selected corrupted data, for more targeted corruptions).


(Other) Weaknesses beyond those mentioned elsewhere:

- [updated after rebuttal] (major) The description of the IB formulation and the variational prior/posterior needs to be made a lot more explicit. I understand the method, but only after the rebuttal. The paper itself needs improvement.
- (minor) Writing needs to be improved around Proposition 4.2: To the reader arriving here, the statement (5) is meaningless, since A and B have not been introduced. Hence, for the reader of the main paper, L_{MI} essentially remains undefined.
- The method is very similar to Subnetwork Probing (SP, Cao et al, cited). The method is compared to, but it is hard to tell which conceptual aspects about the proposed method are responsible for outperforming that method. In particular that the current ms resorts to variational approximations, it is not clear whether the generality of IB is key, or some aspect of the variational approximations. The I[Y;C] term doesn't seem so different from the recoverability-part of the SP objective. It seems the clear conceptual difference lies in the fact that the proposed method uses Gaussian perturbations rather  than hard-concrete random masking, and that the regularization is based on Mutual Information rather than the SP objective. Is this responsible for the performance difference? I think this could in principle be checked in additional experiments. [the authors have provided some new material here]
- The success of the method is only shown on two tasks in a small model (GPT2-small). What are the prospects for scaling the approach to larger models than GPT2Small? Notably, IG-EAG [1] has been evaluated on more tasks and also shown to scale to larger models (Appendix K in [1]). Relatedly, IG-EAG would be good to include as a baseline. [the authors have provided some new material here]

**Questions For Authors:**

I have posed questions about the IB formulation under "Theoretical Claims". Satisfactory clarification could, together with addressing of other weaknesses and concerns, change my evaluation of the paper. [the authors have answered this]

What is the computational cost? It is clear that the proposed method may be more scalable than ACDC, but then it seems much more expensive than (IG-)EAP, depending on the number of iterations needed for training? [the authors have answered this]

**Relation To Broader Scientific Literature:**

The key contributions are to propose a circuit discovery method that
- is based on a holistic differentiable objective
- works even without manually created corrupted activations

**Theoretical Claims:**

I have questions about the basic formulation in terms of IB.
Section 3 defines the circuit and the full model to be computation graphs.
Section 4 introduces the two key information terms based on IB:
 * informativity about true output: I[Y; C]
 * informativity about full graph: I[G; C]

Now a few questions [NOTE after rebuttal: I now understand this. The point remains that the clarity of the paper needs to be very strongly improved in this regard -- I have increased my score trusting the authors that they will keep their promise to revise this aspect thoroughly]:
* (a) In general, I[...;...] is applied to random variables, but here it is not clear where the randomness comes from. Equation (3) has the subscript P(C]G) -- literally, this appears to suggest that the circuit C (i.e., a computation subgraph) itself is a random variable, a random subset of the full graph -- but that would seem at odds with later parts of the paper, so probably is not what is meant. I assume that C here denote activations of the nodes in the graph, so that P(C]G) is a distribution over activation patterns in the circuit, defined by the randomness of the Gaussian perturbations, right?
* (b) Following up, in I[G; C] what is the random variable G? Is it the distribution of activation patterns in the full graph, with the randomness coming from sampling different input strings? But this is actually at odds with Appendix A.2.1, where the KL divergence of two Gaussians is considered. But the activations in G are not Gaussians -- they are deterministic when fixing the input string. Where does Gaussianity come in? Maybe some kind of Gaussianity is imposed also on the full model's activations (which is how I understand Appendix A2), but then this needs to be made very explicit in the main paper.
* (c) Regarding I[Y; C], it remains unclear in Section 4 how this is computed or approximated. Is the idea to construct a variational bound by computing the average (across the randomness in the Gaussian perturbations \epsilon) cross-entropy of the perturbed model on the outputs Y?

---

> ### Author Rebuttal · Authors · 2025-04-01
>
> We sincerely appreciate your effort in reviewing our paper and providing constructive feedback. We would like to address your concerns below and revise the corresponding parts in the revised version.
>
> **Theoretical Claims:** We would like to further summarize the IBCircuit. IBCircuit can be regarded as the application of the Information Bottleneck principle to circuit discovery with our proposed appropriate adaptations.
> * **Why use variational approximation?** The mutual information is intractable and IB can not be directly applied to circuit discovery. The variational method is introduced to estimate the bound of mutual information. The problem is also analyzed in VAE[1] and VIB[2] in deep representation learning.
> * **Random Variable.** In deep neural network[1][2], the input $X$ are random variables. Each concrete input string $x$ can be viewed as an instance sampled from the task distribution $X$, e.g. IOI task. Similarly, $Y$, $\mathcal{C}$ and $\mathcal{G}$ are also random variables, and $[x,y,c,g]$ is an concrete instance. We provide an analysis on random variables, but this formulation can be trained with instances[1][2].
> * **Gaussian in Appendix A.2.** Gaussian is the variational prior[1][2]. Gaussian is widely used as variational priors due to the reparameterization trick and convenient KL calculation[1][2]. Following existing literatures, in IBCircuit, the prior of circuit $\mathcal{C}$ is set as Gaussian. IBCircuit will learn the posterior distribution of $\mathcal{C}$ over task $X$, and we report the mean of the learned distribution $\mathcal{C}$ as the final result.
> * **$I[Y; \mathcal{C}]$ Computation.** For an instance $x$ in a batch, we compute the KL divergence between the output of the perturbed model $y_{\mathcal{c}}$ and the output of the clean model $y$. Then we take the average over the batch. For each instance $x$ in the batch, the corresponding circuit $c$ is generated from Gaussian with reparameterization trick[1].  Therefore, each instance in the training stage is deterministic.
> * **$I[\mathcal{G}; \mathcal{C}]$ Computation in Appendix A.2.** We compute the KL between the posterior distribution and Guassian prior of the circuit over the batch. As the prior is Gaussian[2], we can calculate the KL as in Appendix A.2.
>
> **We will make them clear in the revision.**
>
> [1] Deep Variational Information Bottleneck. ICLR 2017.
>
> [2] Auto-Encoding Variational Bayes. ICLR 2014.
>
> >**W1**: The writing around Proposition 4.2.
>
> Thanks for the comment. We will provide the definitions of A and B in the main text. $L_{MI}(G; C)$ is defined as the upper bound of $I(G; C)$, i.e. the KL between the posterior distribution and Gaussian prior of the circuit over the batch.
>
>
> >**W2**: Comparison to Subnetwork Probing.
>
> * **Whether the IB or the variational approximation is key.** As mentioned above, the key is IB. The variational approximation makes  IB trainable in deep neural networks.
> * **What is responsible for the performance difference?** We think the objective of IBCircuit is responsible for the performance difference. Gaussian perturbations and hard masking are just two different ways of modeling circuits with learnable parameters, not the key. (1) In $I[Y;\mathcal{C}]$, IBCircuit calculate KL while SP calculate cross-entropy. KL measures the distribution of the output logit vector. While cross-entropy only focuses on the ground truth, i.e. one element on the logit vector. SP ignores the logits of other tokens. (2) In $I[\mathcal{G}; \mathcal{C}]$, IBCircuit ensures the circuit does not receive irrelevant information from the whole computation graph. In contrast, SP only encourages sparsity. Even if the circuit obtained by SP is sparse, it may contain some irrelevant information.
>
> >**W3&Theoretical Claims**: Additional Experimental Results.
>
> Thanks for the suggestions. We agree that additional comparisons would enhance our contribution. We include the results about (1) the comparison to IG-EAG, (2) more tasks, and (3) larger models in [this anonymous link](https://anonymous.4open.science/r/IBCircuit_ICML25_rebuttal-76E3/). Due to our limited GPU resources, we have not yet completed all experiments. We will keep the remaining results updated in the next few days.
>
> >**Questions For Authors**: What is the computational cost?
>
> Taking the IOI task as an example, running the official ACDC implementation on a single H20 GPU takes approximately 16 hours, while EAP takes about 1 minute, and the proposed method requires roughly 1.2 hours. Although EAP has a faster runtime, it requires designing corresponding clean and corrupted input pairs to compute the gradients. Moreover, the proposed method outperforms EAP on both IOI and Great-than tasks. Furthermore, our approach is significantly more scalable compared to ACDC.

---

> > ### Comment · Reviewer_tPrr · 2025-04-02
> >
> > I thank the authors for their rebuttal. The authors and me seem to be largely on the same page about these points (including those I mentioned as weaknesses).
> >
> > Thanks for the exposition of the variational approximation, which is helpful. I do understand the role of the Gaussian prior. And the posterior is a factorized Gaussian (independent for each intermediate activation) defined by \lambda_i, \mu_i, \sigma_2^2, right (\mu and \sigma are parameterized via MLPs or something like that -- I didn't find this information anywhere)? I would like to urge the authors to make the presentation of these aspects much more accessible and explicit, in the appendix if needed. That is:
> >
> > * exactly define the random variables going into IB (incl what type of object G and C are -- I gather they are collections of random activations)
> > * exactly define the prior and posterior of the variational approximations
> > * exactly define how \mu_i, \sigma_i are parameterzied
> >
> > I consider making this more explicit an absolute prerequisite for publication. The other reviewers didn't seem to be so concerned about this, but I found the presentation very confusing even though I have published on variational IB methods.
> > > For an instance $x$ in a batch, we compute the KL divergence between the output of the perturbed model $y_c$ and the output of the clean model $y$. Then we take the average over the batch. For each instance in the batch, the corresponding circuit is generated from Gaussian with reparameterization trick[1]. Therefore, each instance in the training stage is deterministic.
> >
> > I am confused about the statement that this KL divergence is computed for an individual instance $x$. First, $y_c$ is not Gaussian and its density seems intractable, because it is a complicated function of the many Gaussian perturbations at individual components transformed by nonlinear components of the transformer. Second, $y$ is not random for any individual $x$. I assume the authors actually compute a cross-entropy loss of Y under the random $y_c$ -- which can be viewed of as a sample-based estimate of D_{KL}(Y||Y_C).
> >
> > Overall, I do stand by my point that clarity about the method needs to be substantially improved in the paper. I will update my review to reflect improved understanding of the paper, but this does not detract from the authors' obligation to improve writing substantially in this regard. I also stand by the other weaknesses I mentioned, which the authors have contextualized and acknowledged in their thoughtful rebuttal.

---

> > > ### Author Response · Authors · 2025-04-09
> > >
> > > Thank you for dedicating your valuable time to engage in the discussion. We would like to address your additional concerns below.
> > >
> > > > **How to parameterize $\mu_i$ and $\sigma_i$?**
> > >
> > > Based on the goal of the circuit discovery, we assume that the circuit $\mathcal{C}$ over the task $X$ follows a distribution. Each instance is sampled from this distribution. Consequently, we directly define the mean and variance of $\mathcal{C}$ as learnable parameters. This aligns with the assumption that the circuit $\mathcal{C}$ is shared across samples of task $X$. We do not learn a different mean and variance for each sample. Therefore, we don’t need an encoder to map each sample to a unique mean and variance. And we report the mean of the learned $\mathcal{C}$ as the final result.
> > >
> > > > **Clarification about the computation of $D_{KL}(Y||Y_C)$.**
> > >
> > > We agree that "$y_c$ is not Gaussian and its density seems intractable" and "$y$ is not random for any individual $x$".  We also agree that "the authors actually compute a cross-entropy loss of $y$ under the random $y_c$". In fact, considering $H(y,y_c)=H(y)+D_{KL}(y||y_c)$ and $H(y)$ is a constant, $D_{KL}(y||y_c)$ and cross entropy $H(y,y_c)$ are equivalent. In our implementation, we use $D_{KL}(y||y_c)$ as objective function.  We set the output vector of clean LLM as $y$, and we do not convert it into hard one-hot label. We want to study the behaviors of LLM over specific task, rather than the point with maximum probability.
> > >
> > > > **Ablation study: Why does IBCircuit outperform SP?**
> > >
> > > To validate the contributions of each component in **IBCircuit**, we conduct the following ablation study:
> > > - **Comparison 1: *Gaussian Perturbations* vs. *Hard-concrete Masking***
> > > - **Comparison 2: *Mutual Information Regularization* vs. *SP Objective***
> > >
> > > The detailed results and analysis are shown in the [anonymous link](https://anonymous.4open.science/r/IBCircuit_ICML25_rebuttal-76E3/Experimental%20Results%20for%20Reviewer%20tPrr/Ablation-Why%20does%20IBCircuit%20outperform%20SP.md).
> > >
> > > >**More tasks, scalability to Larger Models, and comparison with EAP-IG.**
> > >
> > > During the rebuttal period, we supplement the following experiments to demonstrate the effectiveness and scalability of **IBCircuit**:
> > >
> > > (1) Add evaluation on the **Gender-Bias** task.
> > >
> > > (2) Extend comparisons with **EAP-IG** on the **IOI**, **Greaterthan**, and **Gender-Bias** tasks.
> > >
> > > (3) Follow the experimental setup of **EAP-IG** to evaluate **IBCircuit** on the **IOI** task using **GPT-2 XL**.
> > >
> > > The detailed results and analysis are shown in the [anonymous link](https://anonymous.4open.science/r/IBCircuit_ICML25_rebuttal-76E3/Experimental%20Results%20for%20Reviewer%20tPrr/More%20tasks,%20scalability%20to%20Larger%20Models,%20and%20comparison%20with%20EAP-IG.md).
> > >
> > > >**Major Concern: The formulation needs to be made more explicit.**
> > >
> > > Regarding the major concerns, we thank the reviewers for making us aware of the importance of this issue. The ICML does not allow for submitting revisions during the rebuttal period, but we promise to thoroughly address these issues in the final version if our manuscript is accepted. We will also include the additional experimental results during the rebuttal period in the final version. We are very grateful for the reviewers' constructive comments, which improve the quality of our work.

---

### Official Review · Reviewer_eJCi · 2025-03-18

**Overall Recommendation:** 2

**Summary:**

This paper explores circuit discovery in pretrained language models. The proposed method, i.e., IBCircuit, leverages the information bottleneck principle to holistically identify and optimize circuits without needing specific corrupted activations for different tasks. It is demonstrated that IBCircuit can identify more relevant and minimal circuits compared to existing methods, particularly in tasks like Indirect Object Identification and Greater-Than calculations.

**Claims And Evidence:**

Mostly.

**Essential References Not Discussed:**

Not I can think of.

**Experimental Designs Or Analyses:**

Yes.

**Methods And Evaluation Criteria:**

Yes.

**Other Comments Or Suggestions:**

N/A

**Other Strengths And Weaknesses:**

**Strengths:**

- Circuit discovery is a crucial field for enhancing the explainability of language models.
- The paper is well-written and aptly frames the circuit discovery challenge within the Variational Information Bottleneck (VIB) framework.
- The experiments conducted demonstrate the method's effectiveness on a smaller GPT-2 model, providing preliminary evidence of its utility.

**Weaknesses:**

- Eq. 1 introduces the concept of "distorted information flow", which is confusing and not properly explained in the paper. If the goal is to frame this as an Information Bottleneck problem, why is there a need to corrupt the input instead of directly minimizing the KL divergence with the same input? This aspect could benefit from further clarification.

- The proofs presented for the theoretical bounds lack novelty. The derivation of the lower bounds appears to be a straightforward application of previous works.

- It's uncertain whether this method is scalable to larger models like LLAMA, which have been explored in recent studies in circuit detection. Further experimentation on such models would be beneficial to validate the method's applicability across different scales.

**Questions For Authors:**

Please address my concerns in the weakness in "Other Strengths And Weaknesses" section.

**Relation To Broader Scientific Literature:**

Circuit discovery is very important to promote the explainability of black-box pretrained language models

**Theoretical Claims:**

I checked the proof for the VIB.

---

> ### Author Rebuttal · Authors · 2025-04-01
>
> Thank you for your valuable feedback on our paper. We appreciate your time and effort in reviewing our work. We would like to address your questions and concerns below.
>
> >**Weakness 1**: Unclear explanation of "distorted information flow" in Eq.1: Why use input corruption instead of direct KL minimization?
>
> Thanks for the comment. The distorted information flow is used for modeling circuits with learnable parameters. There is no explicit circuit within the LLM, thus we can not directly minimize KL. In the distorted information flow, we model the learnable circuit with Gaussian, i.e. input corruption. Therefore, IBCircuit can optimize the circuit with gradient descent. It is the common practice in variational methods [1][2]. Using input corruption to model circuit is common in optimization framework for circuit discovery. For example, the baseline Subnetwork Probing [3] uses a hard-concrete random masking as input corruption to model the circuit.
>
> [1] Deep Variational Information Bottleneck. ICLR 2017.
>
>
> [2] Auto-Encoding Variational Bayes. ICLR 2014.
>
> [3] Low-complexity probing via finding subnetworks. NAACL 2021.
>
> >**Weakness 2**: The proofs presented for the theoretical bounds lack novelty. The derivation of the lower bounds appears to be a straightforward application of previous works.
>
> Thanks for the comment. We derived a novel upper bound, and since previous subgraph discovery work has the same optimization objective as circuit discovery, we adopted the same form of lower bound.
>
> * **Upper Bound:** The traditional upper bound can not be directly applied to the circuit discovery, as the random variables $\mathcal{G}$ and $\mathcal{C}$ are computational graphs. Therefore, we derive a new upper bound.
>
> * **Lower Bound:** In previous studies, the lower bound derived for subgraph discovery aims to identify critical subgraphs that preserve information similar to the complete graph. This aligns with the objective of circuit discovery, where the goal is to find circuits that maintain the performance of the full pre-trained model. Therefore, we incorporate the same bound using the notation of Information Bottleneck Circuit to ensure self-consistency in the proposed method.
>
> >**Weakness 3**: No evidence for applicability to larger models.
>
> Thanks for the suggestions. We agree that additional results on larger models would enhance our contribution. We include the results in [this anonymous link](https://anonymous.4open.science/r/IBCircuit_ICML25_rebuttal-76E3/). Due to our limited GPU resources, we have not yet completed all experiments. We will keep the remaining results updated in the next few days.

---

### Decision · Program_Chairs · 2025-05-01

**Decision:**

Accept (poster)

**Comment:**

**Summary**

This paper proposes IBCircuit, a novel approach for circuit discovery in language models based on the Information Bottleneck principle. Unlike previous methods that require manually designed corrupted activations, IBCircuit provides an end-to-end optimization framework that can be applied to any task without tedious corrupted activation design. The authors demonstrate that IBCircuit identifies more faithful and minimal circuits in terms of critical node components and edge components compared to recent related work on the Indirect Object Identification (IOI) and Greater-Than tasks. The method shows promising results in terms of faithfulness metrics and scaling potential, offering a new direction for interpretability research in large language models.

**Reasons to Accept**

- Circuit discovery is a crucial field for enhancing the explainability of language models (eJCi, tPrr)
- The method is competitive or even better than existing methods in terms of AUROC and faithfulness metrics (tPrr, vc6F, ZTi2)
- The idea of circuit discovery based on end-to-end optimization is very appealing and more scalable than iterative methods like ACDC (tPrr)
- The method is significantly faster than ACDC, taking about 1.2 hours compared to 16 hours for ACDC on the IOI task using the same hardware (ZTi2)

**Suggestions for revisions**
1. The formulation and variational implementation of the Information Bottleneck approach is not sufficiently explained in the paper, making it difficult for readers to understand how the method works (tPrr)
2. The method has only been evaluated on a smaller GPT-2 model and limited tasks (IOI and Greater-Than), raising questions about its scalability to larger models like LLAMA (eJCi, tPrr, vc6F)
3. The proofs for theoretical bounds lack novelty, appearing as straightforward applications of previous works (eJCi)
4. Missing comparisons with recent scalable circuit discovery methods like Edge Pruning and IG-EAG (vc6F, tPrr)
5. Equation 1 introduces "distorted information flow" without proper explanation, raising questions about why input corruption is needed instead of direct KL minimization (eJCi)
6. The evaluation treats previous manually validated circuits as "ground truth," which may unfairly penalize methods that discover valid components not identified in those circuits (ZTi2)
7. Using KL Divergence as a faithfulness metric may recover more than just task-specific behavior (ZTi2)
8. The paper misses citation of several relevant works, particularly regarding information-theoretic approaches to circuit discovery (ZTi2)